# Hair follicle stem cells regulate retinoid metabolism to maintain the self-renewal niche for melanocyte stem cells

Zhiwei Lu[1,2], Yuhua Xie[2], Huanwei Huang[2], Kaiju Jiang[2], Bin Zhou[3], Fengchao Wang[2], Ting Chen[2,4]*

[1]Peking Union Medical College, Beijing, China; [2]National Institute of Biological Sciences, Beijing, China; [3]Institute of Biochemistry and Cell Biology, Shanghai Institutes for Biological Sciences, University of Chinese Academy of Sciences, Beijing, China; [4]Tsinghua Institute of Multidisciplinary Biomedical Research, Tsinghua University, Beijing, China

**Abstract** Metabolites are major biological parameters sensed by many cell types in vivo, whether they function as signaling mediators of SC and niche cross talk to regulate tissue regeneration is largely unknown. We show here that deletion of the Notch pathway co-factor RBP-J specifically in mouse HFSCs triggers adjacent McSCs to precociously differentiate in their shared niche. Transcriptome screen and in vivo functional studies revealed that the elevated level of retinoic acid (RA) caused by de-repression of RA metabolic process genes as a result of RBP-J deletion in HFSCs triggers ectopic McSCs differentiation in the niche. Mechanistically the increased level of RA sensitizes McSCs to differentiation signal KIT-ligand by increasing its c-Kit receptor protein level in vivo. Using genetic approach, we further pinpointed HFSCs as the source of KIT-ligand in the niche. We discover that HFSCs regulate the metabolite RA level in vivo to allow self-renewal of neighboring McSCs.

*For correspondence: chenting@nibs.ac.cn

Competing interests: The authors declare that no competing interests exist.

## Introduction

The concept of a 'SC niche' was first proposed by *Schofield (1978)*, even though there was no direct evidence that such a niche actually existed at that time. Since then, large strides have been made towards understanding the cellular and molecular composition of SC niches in multiple experimental systems (*Crane et al., 2017*; *Sailaja et al., 2016*; *Chen et al., 2016*). Now it's widely accepted that niche space plays dominant roles in SC fate establishment, maintenance, activation and SC related disease initiation (*Xu et al., 2015*; *Scadden, 2007*; *Clevers et al., 2014*). Recently several studies have revealed that SCs themselves actively organize the very niche space they occupy in vivo. For example during early zebra fish development, the arrival of hematopoietic SCs triggers dynamic remodeling of the perivascular niche (*Tamplin et al., 2015*). In medaka, neural SCs induce the formation of their physical niche during organogenesis (*Seleit et al., 2017*). However, less is known about the role SCs might play in regulating the niche signaling environment, beyond their demonstrated capacity to passively receive signaling inputs.

Mouse dorsal skin HFs regenerate continuously. Their cyclical bouts of growth (anagen), regression (catagen) and rest (telogen) are driven by the proliferation and stepwise differentiation of HFSCs residing in the bulge area. McSCs also reside in the bulge, and the secondary hair germ (sHG) located below the bulge (*Nishimura et al., 2002*). Both types of SCs remain undifferentiated in their shared niche space and are coordinately activated at the onset of anagen to generate a pigmented hair shaft. Studies have identified numerous signaling inputs received by HFSCs and McSCs respectively or collectively in their shared niche space. For example, BMPs and FGF18 are essential

to maintain HFSCs in a dormant state (*Hsu et al., 2011*). TGF-b signaling is required to suppress McSC proliferation and differentiation, while active Notch signaling via Hes1 is required for McSCs survival (*Moriyama et al., 2006*; *Nishimura et al., 2010*). At the onset of anagen, the dynamic changes of multiple signaling molecules activate hair regeneration (*Greco et al., 2009*). Among them, Wnt signaling coordinates the activation of both HFSCs and McSCs (*Rabbani et al., 2011*); KIT-ligand is necessary for McSC mobility and differentiation (*Nishikawa et al., 1991*); Nfib/Edn signaling can regulate the differentiation of McSCs in collaboration with KIT or Wnt signaling (*Chang et al., 2013*; *Takeo et al., 2016*). Compared to these well-studied morphogens, whether metabolites can influence SC fate during tissue regeneration in vivo is still largely unknown.

Retinoic acid (RA) is the major product of carotenoid oxidation pathway and it can also be transformed from Vitamin A (*Marill et al., 2003*). As a small molecule, RA can diffuse freely through cell membrane and is essential for normal regulation of a wide range of biological processes including development, differentiation, proliferation and apoptosis (*Zile, 2004*; *Li et al., 2012*; *Glover et al., 2006*; *Maden, 2006*; *Kelley et al., 1994*; *Niederreither et al., 2001*). In skin, RA has long been used as additive in makeup or acne cream (*Orfanos et al., 1987*). There are lots of studies about RA and keratinocytes or fibroblasts. In makeup, RA is used to increase collagen synthesis (*Schwartz et al., 1990*) and reduce keratinocyte differentiation (*Yaar et al., 1981*). In acne cream, RA acts mainly as comedolytics, but anti-inflammatory actions have also been suggested (*Chivot, 2005*). However, there are much fewer reports about how RA regulates melanocytes, especially in vivo.

RBP-J is a Notch pathway co-factor. When Notch signaling is inactive, RBP-J binds its target genes and recruits repressive transcription factors (TFs) that silence gene transcription. Once a Notch ligand (either Dll1-4 or Jag1-2) binds to a Notch receptor (mainly Notch1-3 in skin), the receptor is then cleaved, and the intracellular domain (ICD) translocates into the nucleus and binds to RBP-J. Subsequently, instead of recruiting repressive TFs, the complex starts to activate the transcription of downstream genes (*Borggrefe and Oswald, 2009*). Previously, it has been reported that Notch pathway is essential for skin epidermal cell differentiation, HF formation, dermal papillar niche function and crosstalk between the epidermis and the HF (*Lin et al., 2000*; *Kopan et al., 2002*; *Lee et al., 2007*; *Rangarajan et al., 2001*; *Blanpain et al., 2006*; *Hu et al., 2010*).

Unexpectedly, we found here that during normal homeostasis, the Notch pathway co-factor RBP-J functions in HFSCs to suppress retinoid metabolic process genes. The increased level of RA resulted from loss of RBP-J in HFSCs elevates the c-Kit protein level on McSCs and sensitizes them towards KIT-ligand normally present in the niche. This reveals that beyond the SCs demonstrated capacity to passively receive signaling inputs, they also play an essential role of safeguarding the differentiation refractory niche environment by regulating metabolite level in vivo. The fact RA is the messenger mediating the crosstalk between two types of SCs in a shared niche, reveals the previously unknown role of metabolites in regulating SC function during tissue regeneration and possibly disease development in vivo.

## Results

### Specific ablation of RBP-J in HFSCs results in ectopic differentiation of neighboring McSCs in the shared niche

Our previous studies discovered that the niche environment plays a dominant role in HFSC fate establishment during morphogenesis (*Xu et al., 2015*). Since maintaining the undifferentiated state is a defining feature of SCs, we decided to address the key question of how niche helps maintain SCs stay undifferentiated. Based on our previous transcriptome and signaling pathway analysis (*Xu et al., 2015*), we used *Sox9-CreER* to conditionally knock out (cKO) the canonical Notch pathway co-factor gene *Rbpj*. First the labeling specificity of *Sox9-CreER* was determined by using the *Sox9-CreER::Ai14* mice. Ai14 allele was used to mark all *Sox9-CreER* expressing cells as RFP+. Tamoxifen treatment from postnatal (P) day 1 to 4 results in specific labeling of HF epithelial cells including the HFSCs, but not the McSCs (*Figure 1A*). Efficient *RBP-J* ablation by *Sox9-CreER::Rbpj <fl/fl>* in HF epithelial cells was confirmed by RBP-J staining at the first telogen (*Figure 1B*).

Loss of RBP-J in HF epithelial cells does not lead to immediate loss of HFSC markers CD34 and Sox9 (*Figure 1B* and *Figure 1—figure supplement 1A*), nor does the overall morphology of the

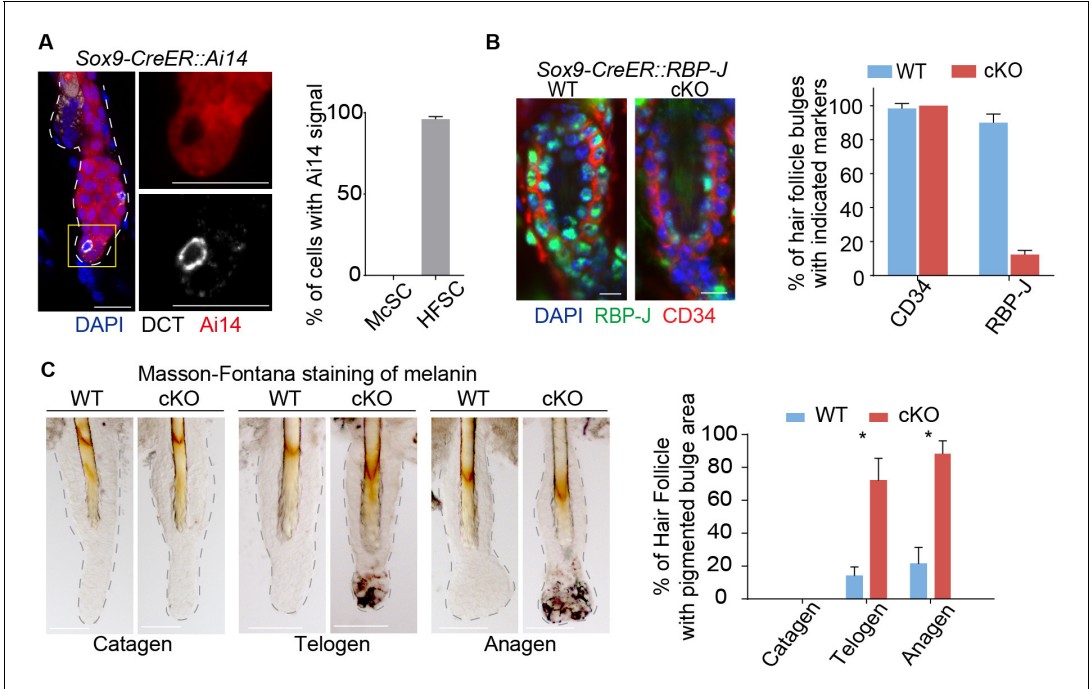

**Figure 1.** Ablation of Notch pathway effector RBP-J in bulge epithelial cells causes McSCs ectopic differentiation in the niche. (**A**) Representative immunofluorescence images and quantification of Tamoxifen induced *Sox9-CreER::Ai14* mice indicating efficient labeling of bulge epithelial cells but not McSCs. DCT is a melanocyte marker. Tamoxifen was injected on P1-4 at anagen, dorsal skin samples were taken on P20 at telogen. (**B**) Representative immunofluorescence images and quantification of CD34 and RBP-J in the bulge of *WT* and *Sox9-CreER::RBP-J cKO* HFs in dorsal skin. Note the efficient ablation of RBP-J in both HFSCs (marked by CD34) and the inner layer CPLs in *cKO* compared to *WT* bulge. (**C**) Representative tail skin wholemount melanin specific Masson-Fontana staining images and quantification of ectopic pigmentation in the bulge of *WT* and *Sox9-CreER:: RBP-J cKO* HFs at the telogen to anagen transition stages. Tamoxifen was injected on P1-4 at anagen, tail skin samples were taken on P14(catagen), P15(telogen) and P16(anagen). All data are expressed as mean ± SD ≥ 20 follicles are quantified each mouse. N = 3 at each time point. (*) p<0.05. Scale bars = 10 μm.

The online version of this article includes the following figure supplement(s) for figure 1:

**Figure supplement 1.** HF phenotype in *Sox9-CreER::RBP-J cKO* mice.

telogen bulge change. But unexpectedly, we noticed the bulge region in the *Sox9-CreER::RBP-J cKO* HFs show ectopic pigmentation at the telogen to anagen transition stage, which is not observed in the *WT* HFs (**Figure 1C** and **Figure 1—figure supplement 1B,C**). This is very peculiar because the McSCs, which also reside in the bulge region, are supposed to be undifferentiated, and only their downstream progenies located in the lower HF bulb region undergo terminal differentiation to help generate pigmented hair shaft during anagen. Based on melanin specific Masson-Fontana staining and quantification, more than 60% of the *Sox9-CreER::RBP-J cKO* HFs start to show ectopic pigmentation in the bulge region at telogen to anagen transition stage. This is even more obvious in early anagen when about 90% of the bulge become pigmented, in comparison less than 20% of the *WT* HFs show pigmentation in the bulge region (**Figure 1C**).

This hair cycle-dependent ectopic differentiation of McSCs revealed a crosstalk between the HF epithelial cells with McSCs. But the broad expression pattern of *Sox9-CreER* in all HF epithelial cells cannot pinpoint the specific responsible cell type for this phenotype. Additionally, we also observed aberrant terminal differentiation and total degeneration of the HF structure in the first anagen after morphogenesis ablation of *RBP-J* in *Sox9-CreER::RBP-J* mice (**Figure 1—figure supplement 1D,E**). This is consistent with previous publication that as the main co-factor of the canonical Notch pathway RBP-J is required for HFSC downstream progenies terminal differentiation (**Demehri and Kopan, 2009**). The degenerated HF structure prevents further dissection of the underlying cellular and molecular mechanisms mediating the crosstalk between epithelial cells and McSCs in the shared niche environment.

To solve these problems, we generated a *Krt6-CreER* knockin mouse line to specifically KO *RBP-J* in telogen CD34+ HFSCs but not the sHG cells, which proliferate and generate downstream progenies undergoing differentiation in lower anagen HFs (*Greco et al., 2009*). To this end, we used CRISPR/Cas9 to insert the *IRES-CreER* cassette after the *Krt6a* exon9 (*Figure 2A*). The resulting successful insertion was confirmed by genotyping and sequencing. The labeling specificity of *Krt6-CreER* was determined by using the *Krt6-CreER::Ai6* mice. Ai6 allele marks all the *Krt6-CreER* expressing cells as GFP+. When tamoxifen was injected during the second telogen, we found that CD34+ HFSCs and Krt6+ companion layer cells (CPL) are efficiently labeled in close to 100% of the HFs, but only less than 10% of HFs contain occasionally labeled cells in the sHG (*Figure 2B* and *Figure 2—figure supplement 1A*). FACS analysis indicates that ~80% of HFSCs in dorsal skin are labeled using our *Krt6-CreER* line (*Figure 2—figure supplement 1B*).

To specifically ablate *RBP-J* in bulge cells but not sHG cells, *Krt6-CreER::RBP-J* mice were treated with tamoxifen during the second postnatal telogen, which lasts several weeks (*Figure 2—figure supplement 1C*). Restricted loss of *RBP-J* in HFSCs and CPL cells but not sHG cells was confirmed by RBP-J staining at telogen (*Figure 2C*, *Figure 2—figure supplement 1D*). When cKO follicles enter anagen, cells in the lower HF generated by telogen sHGs cells are all RBP-J positive, while the upper HF bulge cells are all RBP-J negative (*Figure 2D*). Consequently, anagen follicles in the first round are normal in the *Krt6-CreER::RBP-J* mice. This allows us to carry out in-depth cellular and molecular study of the McSC phenotype observed. McSCs do not express *Krt6* and the number of McSCs in the bulge and their downstream progenies located in the bulb of anagen HFs are normal following telogen cKO of RBP-J in *Krt6-CreER::RBP-J* mice (*Figure 2—figure supplement 1E*). Consistent with the observed hair cycle dependent bulge pigmentation phenotype in *Sox9-CreER::RBP-J* mice, we did not observe ectopic pigmentation or McSC abnormal differentiation in the telogen bulge after RBP-J ablation in *Krt6-CreER::RBP-J* mice (*Figure 2—figure supplement 1G,H*). Only when hair follicles spontaneously enter anagen can pigmentation be observed in cKO HFs (*Figure 2—figure supplement 1I*). Because the *RBP-J cKO* HFSCs generate the sHG cells and the eventual downstream progenies in later rounds of hair cycles, we observed abnormal terminal differentiation and total degeneration of the HF structure in *Krt6-CreER::RBP-J* mice at later time points (*Figure 2—figure supplement 1F*). The eventual HF degeneration in mutant skin precludes the long-term observation of McSC ectopic differentiation phenotype.

To pinpoint if *RBP-J* null HFSCs are solely responsible for causing the McSCs ectopic differentiation phenotype, we used wax to remove hair shaft and the attached CPL cells (*Figure 2E*). HF wax at telogen only removes the club hair and the attached Krt6+ CPL cells but not the HFSCs (*Figure 2F*). Consequently, the only *RBP-J cKO* cells left in HF are the HFSCs, and the *RBP-J cKO* CPL cells are removed from the equation. Removal of CPL cells at telogen also triggers anagen entry (*Chen et al., 2012*) (*Figure 2G*). This allows us to follow the hair cycle dependent ectopic differentiation of McSCs in a temporally controlled fashion.

Cell apoptosis and proliferation are the same in bulge area between WT and cKO follicles (*Figure 2H* and *Figure 2—figure supplement 1J*). After HF wax induced anagen entry, the *cKO* HFs start to show ectopic pigmented in bulge at the telogen to anagen transition stage (*Figure 2I,J*). Starting at D3 post wax, 90% of the *cKO* HFs contain pigmented bulge, compared to less than 20% of *WT* HFs with the same phenomenon (*Figure 2I*). We also detect increased Tyrp1 levels in McSCs of the *cKO* HFs starting at D2 post wax when the HFs are still in telogen, while in *WT* HFs the expression of Tyrp1 is restricted to differentiated melanocytes located in the bulb of anagen HFs (*Figure 2J*). Since the phenotype occurs after removal of the CPL cells, we conclude that McSCs are influenced in a paracrine fashion by loss of *RBP-J* specifically in HFSC. It reveals an essential role of HFSCs in maintaining a differentiation refractory niche environment for the neighboring McSCs.

## RBP-J functions as a repressor in HFSCs to suppress retinoid metabolic process genes

To understand the molecular mechanism mediating the crosstalk between HFSCs and McSCs in a shared niche, we first characterized the expression pattern of RBP-J and its Notch pathway co-factors in HFs. We found RBP-J is expressed by most cells in HFs from the beginning of morphogenesis to adult telogen (*Figure 3A*). In contrast, the active form of Notch1 ICD (N1ICD) is only detected in the terminally differentiated cells in the pre-cortex cells of anagen HFs or CPL cells in telogen HFs, but not in the HFSCs (*Figure 3B*). To examine the expression patterns of other Notch receptor ICDs

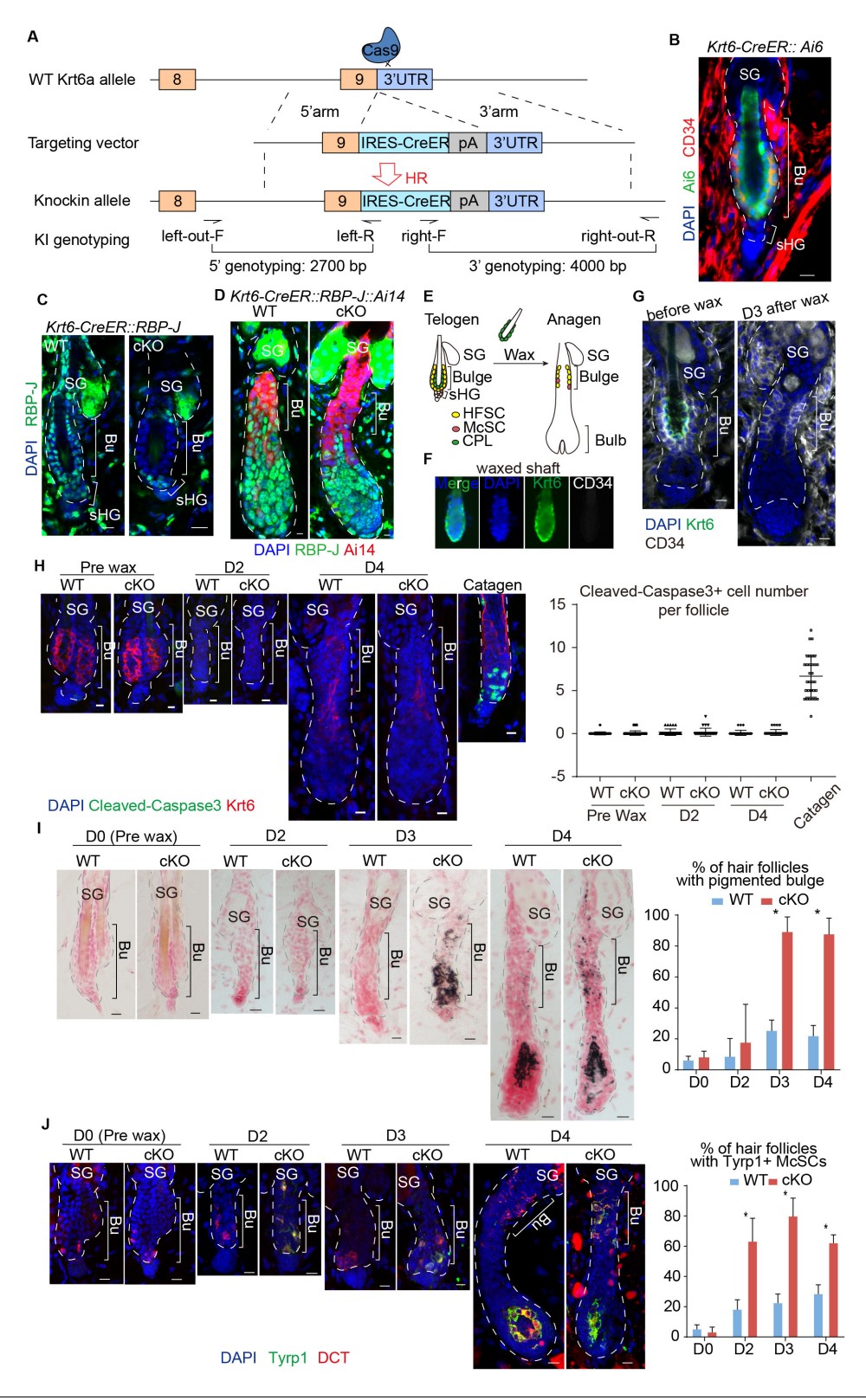

**Figure 2.** Loss of RBP-J specifically in HFSC causes hair cycle dependent McSCs ectopic differentiation in the shared niche. (**A**) Schematic diagram of CRISPR/Cas9-mediated knockin of *CreER* in the 3′ UTR of *Krt6a* locus. sgRNA targets downstream of *Krt6a* stop codon. Two pair of indicated primers spanning the junctions of both homologous arms and the target sequences were used to validate correct insertion. (**B**) Representative immunofluorescence image of *Krt6-CreER::Ai6* labeling HFSCs in telogen bulge. Tamoxifen was injected on P55 at the second telogen, dorsal skin

*Figure 2 continued on next page*

Figure 2 continued

samples were taken two days later at telogen for analysis. Note the high labeling efficiency of both HFSCs (marked by CD34) and the inner layer CPLs in telogen bulge. SG, sebaceous gland; Bu, bulge; sHG, secondary hair germ. (C) Representative immunofluorescence images of RBP-J in telogen bulge of *WT* and *Krt6-CreER::RBP-J cKO* HFs. Tamoxifen was injected on P50-52 at telogen, samples were taken on P58 at telogen. Note the efficient ablation of RBP-J in both layers of cells in the bulge of *cKO* compared to *WT* HF. (D) Representative immunofluorescence images of RBP-J in anagen bulge of *WT* and *Krt6-CreER::RBP-J::Ai14 cKO* HFs. Note in all RFP+ cells RBP-J is ablated, while matrix is not affected. (E) Schematic diagram of wax induced removal of CPL cells and subsequent telogen to anagen transition. (F) Representative immunofluorescence images of waxed hair shaft. Note the presence of Krt6+ CPL cells, but not the CD34+ HFSCs. (G) Representative immunofluorescence images of HFs before and at 3 days (D3) after wax. Krt6+ CPL cells are completely removed from the HF after wax, but CD34+ HFSCs remain intact. At D3 after wax HFs have entered anagen. (H) Representative immunofluorescence images and quantification of Cleaved-Caspase3 at indicated time points of before and after wax. Catagen follicle is used as positive control for staining. (I) Representative dorsal skin Masson-Fontana staining images and quantification of ectopic pigmentation in the bulge of *WT* and *Krt6-CreER::RBP-J cKO* HFs at the telogen to anagen transition stages. Tamoxifen was injected on P50-52 at telogen, then CPL cells were removed by wax on P54 to induce anagen entry. Dorsal skin samples were taken at indicated time points of before (D0) and after (D2/3/4) wax. HFs were counter-stained by neutral red. Brackets indicate the bulge region where McSC ectopic differentiation is quantified. (J) Representative immunofluorescence images and quantification of percentage of HF with Tyrp1+ McSCs. McSC located in the bulge area indicated by brackets were labeled with DCT staining. Melanocyte differentiation marker Tyrp1 in DCT+ McSCs located in the bulge indicates ectopic differentiation. Percentage of HFs with Tyrp1+ McSCs in *WT* and *Krt6-CreER::RBP-J cKO* HFs of dorsal skin were quantified. All data are expressed as mean ± SD ≥ 15 follicles are quantified each mouse. N = 3. Scale bars = 10 μm.

The online version of this article includes the following figure supplement(s) for figure 2:

**Figure supplement 1.** HF phenotype in *Krt6-CreER::RBP-J cKO* mice.

and active canonical Notch pathway downstream genes, we used FACS to isolate HFSCs and the differentiated matrix cells of anagen HFs using *Nfatc1-CreER::Ai14::Krt14H2BGFP* mice (*Figure 3—figure supplement 1A–C*). Tamoxifen was injected at P12 to label HFSCs as Ai14+ cells in upper HFs. HFSCs are isolated as Sca1-GFP+RFP+ cells. Sca1-GFP$^{low}$ cells are sorted as matrix cells due to their frequent cell division that dilutes H2BGFP signals. Specificity of the isolated cells were confirmed by marker expression using Q-PCR: *Nfatc1* and *Sox9* for HFSCs; *SHH*, *Hoxc13*, *Axin2* and *Lef1* for matrix cells (*Figure 3D*). Western blot analysis of these sorted cells revealed little or no expression of N1ICD and N2ICD in the HFSCs, compared to high level of these two proteins in the matrix cells (*Figure 3C*). No N3ICD was detected in either cell population. We then checked the expression pattern of canonical Notch downstream target genes *Hes1* and *Hey1* using qPCR. Consistent with our western blot results, they are much lower in HFSCs compared to differentiated matrix cells (*Figure 3D*). In addition to anagen HFs, we also used FACS to isolate HFSC and the differentiated CPL cells from telogen HFs using *Krt6-CreER::Ai6* mice (*Figure 3—figure supplement 1D,E*). Similarly, active Notch receptor ICDs and canonical Notch downstream targets are lower in HFSCs compared to the differentiated CPL cells (*Figure 3E,F*). All these results indicate active Notch signaling is absent in HFSCs and RBP-J serves as a repressor in them.

We next used FACS to isolate *WT* and *cKO* HFSCs from *Krt6-CreER::RBP-J* mice at D2 after wax to profile their transcriptome difference using RNA-seq. Since RBP-J serves as a repressor in HFSCs, we mainly focused on genes that were up regulated >1.5 fold (p<0.05) in *cKO* compared to *WT* HFSCs (*Supplementary file 1*). Based on our Gene Ontology_Biological Process (GO_BP) analysis of RNA-seq results and previous known factors that regulate melanocyte differentiation, we systematically examined the significantly enriched biological processes or signal pathways (*Figure 3G*). The top category of our GO_BP analysis is lipid metabolic process. Using maker staining or western blot analysis we found no difference in lipid deposition or organelles with lipid rich membrane structures between *WT* and *Krt6-CreER::RBP-J cKO* HFs (*Figure 3—figure supplement 2A–E*). Another highly ranked category is positive regulation of cell death. Since cell death and DNA damage have previously been reported to cause melanocyte differentiation (*Inomata et al., 2009*), we examined different forms of cell death and DNA damage commonly related to cell death. We found no difference in sensitive DNA damage marker $\gamma H_2Ax$, apoptosis marker cleaved Caspase3, or necrosis marker p-MLKL between *WT* and *cKO* HFs (*Figure 3—figure supplement 2F–H*).

The hair cycle dependent ectopic differentiation of McSCs indicates signals involved in telogen to anagen transition might play a role. Since Wnt signaling has been reported to mediate both HFSCs and McSCs activation at this stage (*Rabbani et al., 2011*; *Chang et al., 2013*), we specifically checked the level of Wnt signaling. RNA-seq results showed that Wnt3(~5X) and Axin2(~1.5X) were

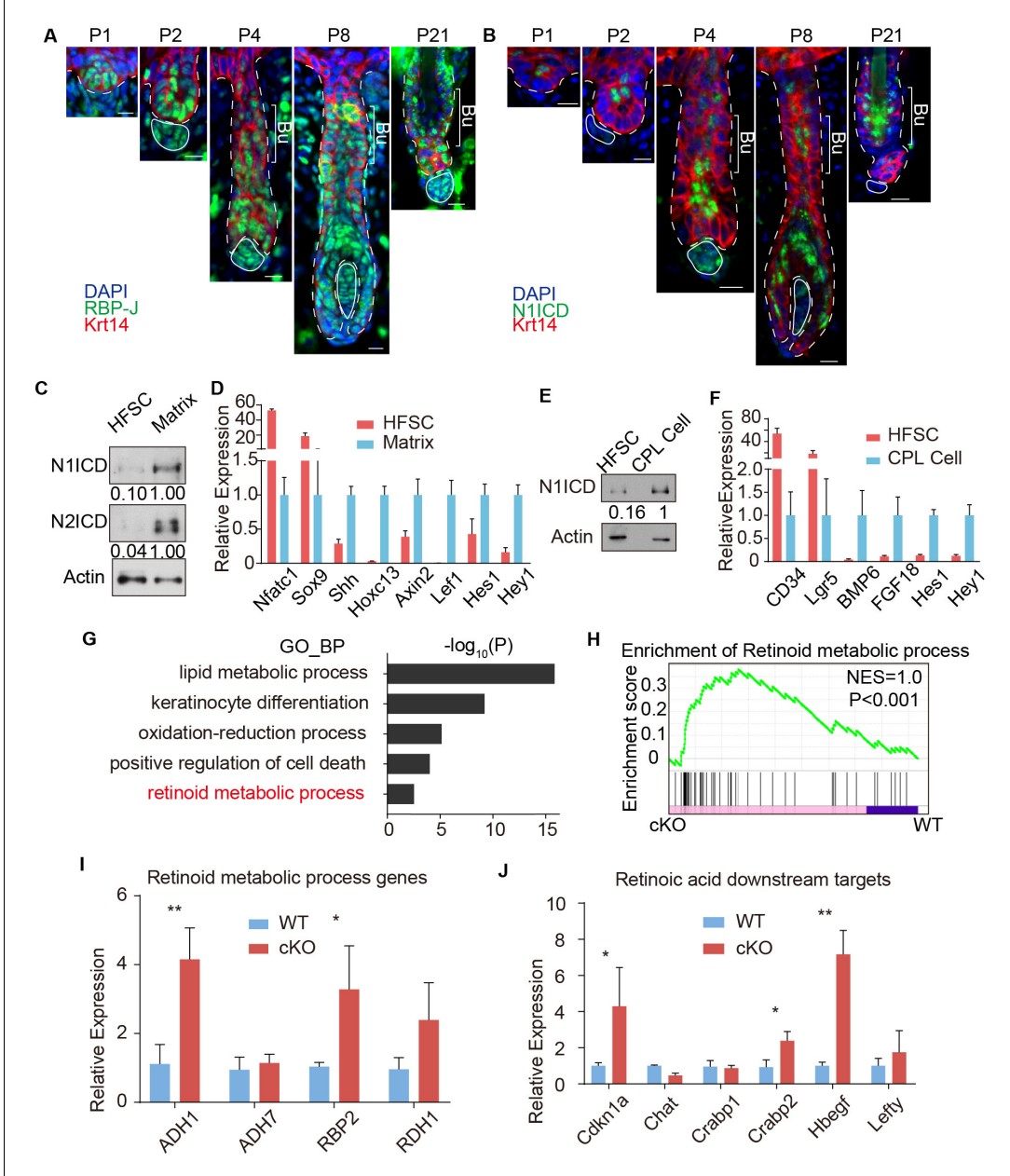

**Figure 3.** RBP-J functions as a repressor in HFSC and downregulates retinoid metabolic process genes. (A) Representative immunofluorescence images of RBP-J demonstrate broad expression pattern in HF epithelial cells at all stages of hair cycle from morphogenesis to adult. Krt14 is a basal epithelial cell marker. (B) Representative immunofluorescence images of Notch1 intracellular domain (N1ICD) show restricted pattern in differentiated cells instead of HFSCs at different hair cycle stages. N1ICD activates canonical Notch signaling. (C) Western blot of N1ICD and N2ICD in FACS isolated HFSCs and differentiated matrix cells from anagen HFs. Expression levels were normalized to Actin. (D) QPCR analysis of active canonical Notch signaling downstream genes *Hes1* and *Hey1* in FACS isolated HFSC and matrix cells from anagen HFs. HFSC marker genes *Nfatc1* and *Sox9*; matrix cell marker genes *Shh, Hoxc13, Axin2 and Lef1* were used to validate the correct cell populations used for analysis. (E) Western blot of N1ICD in FACS isolated HFSCs and CPL cells from telogen bulge. Expression levels were normalized to Actin. (F) QPCR analysis of active canonical Notch signaling downstream genes *Hes1* and *Hey1* in FACS isolated HFSC and CPL cells from telogen bulge. HFSC marker genes *CD34* and *Lgr5*; CPL cell marker genes *BMP6 and FGF18* were used to validate the correct cell populations used for analysis. (G) Gene Ontology analysis of upregulated genes in HFSCs of *Krt6-CreER::RBP-J cKO* compared to *WT*. HFSCs were isolated using FACS from dorsal skin at D2 after wax. Retinoid metabolic process category is marked in red. (H) GSEA results showing enrichment of retinoid metabolic process genes in HFSCs of *Krt6-CreER::RBP-J cKO* compared to *WT*. (I) QPCR analysis of retinoid metabolic process genes in FACS isolated HFSCs from *WT* and *Krt6-CreER::RBP-J cKO* HFs at D2 after wax. (J) QPCR analysis of retinoid downstream targets genes in FACS isolated HFSCs from *WT* and *Krt6-CreER::RBP-J cKO* HFs at D2 after wax. All data are expressed as mean ± SD. N ≥ 3. (*) p<0.05; (**) p<0.01. Scale bars = 10 μm.
*Figure 3 continued on next page*

*Figure 3 continued*

The online version of this article includes the following figure supplement(s) for figure 3:

**Figure supplement 1.** FACS sorting strategy for HFSCs, CPL cells and matrix cells.

**Figure supplement 2.** Signal pathways that do not affect McSCs ectopic differentiation triggered by RBP-J loss in HFSCs.

up-regulated in *cKO* HFSCs compared to *WT* (*Supplementary file 1*). We also found significant increases in the level of nuclear β-catenin staining in McSCs of *cKO* HFs from *Krt6-CreER::RBP-J* mice compared to *WT*, indicating higher Wnt signaling levels (*Figure 3—figure supplement 2I*). To functionally test the relevance of Wnt signaling, we used the Wnt inhibitor LGK974 to inhibit the total Wnt level in the niche (*Figure 3—figure supplement 2I*). We found after LGK974 administration, hair cycle was arrested in telogen and active Wnt signal marker nuclear β-catenin was dramatically decreased (*Figure 3—figure supplement 2I*), indicating efficient blocking of Wnt signaling. However, ectopic differentiation of the McSCs still persisted in the *cKO* HFs after LGK974 treatment (*Figure 3—figure supplement 2J*). The number of McSCs also remains the same in *WT* and *cKO* telogen bulge after LGK974 treatment (*Figure 3—figure supplement 2K*). These results rule out elevated Wnt signaling as the reason behind the observed phenotype.

Finally our GO_BP analysis revealed retinoid metabolic process genes as an unexpected category up regulated in *cKO* HFSCs compared to *WT* (*Figure 3G*). Gene set enrichment analysis also showed differences in retinoid metabolic process related genes between *WT* and *cKO* HFSCs (*Figure 3H*). This result is confirmed by qPCR using FACS isolated HFSCs from *cKO* and *WT* telogen skin. Retinoid metabolic process genes such as *ADH1, RBP2* and *RDH1* are expressed at 3 ~ 4 fold higher in HFSCs from *Krt6-CreER::RBP-J cKO* compared to *WT* HFs (*Figure 3I*). To examine whether RA level is indeed higher in the *cKO* niche compared to *WT*, first we used qPCR to examine the expression level of RA downstream genes in FACS isolated HFSCs. Several of them such as *Cdkn1a, Crabp2, Hbegf,* and *Lefty* are statistically significantly higher in *RBP-J* cKO compared to *WT* HFSCs (*Figure 3J*). We also measured the level of RA directly using ELISA assay. *WT* HFs contain ~50 pg RA per million HFSCs, while the *cKO* HFs contain ~90 pg RA per million HFSCs (Figure 5B). These results revealed that RBP-J functions as a repressor in HFSCs to suppress retinoid metabolic process genes and consequently the level of RA in the niche shared by HFSCs and McSCs.

## Chromatin states of target genes are changed in RBP-J cKO HFSCs

To delve deeper into the mechanism of how loss of RBP-J affects retinoid metabolic process genes expression in HFSCs, we performed Assay for Transposase-Accessible Chromatin using sequencing (ATAC-seq) using FACS isolated HFSCs from *WT* and *Krt6-CreER::RBP-J cKO* HFs. About 6000 in 95,000 peaks were more accessible in *cKO* HFSCs compared to *WT* HFSCs (*Figure 4A*). Relative to the mouse genome, regions with similar accessibility in *WT* and *cKO* HFSCs were enriched at 5'-UTRs and exons. In contrast, regions with more accessibility in *cKO* HFSCs compared to *WT* were enriched for promoter, which fits putative RBP-J binding regions (*Figure 4B,C*).

Next we examined closely at specific genes, particularly those that demonstrate increased expression in *cKO* HFSCs compared to *WT*. We found for canonical Notch signal downstream gene *Hes1*, keratinocyte differentiation gene *Krt10*, and retinoid metabolic process gene *Adh1*, their promoters and 5'-UTRs were more accessible in *cKO* HFSCs compared to *WT* (*Figure 4D*). The DNA regions that demonstrate changed accessibility also overlap with putative RBP-J binding motif based on published RBP-J CHIP-seq analysis (*Meredith et al., 2013*; *Castel et al., 2013*). In contrast, genes that do not show differential expression such as HFSC marker *CD34* demonstrates similar accessibility in its promoter, 5'-UTR and the whole gene region in *cKO* HFSCs compared to *WT*. To quantify these differences, we used statistical analysis of normalized ATAC-seq reads from 3.5 kb upstream to 3.5 kb downstream of indicated genes between *WT* and *cKO*: differences in ATAC-seq reads were observed in *Adh1, Krt10* and *Hes1*, but not in *CD34* (*Figure 4E*).

To directly test whether *RBP-J* functionally suppress the expression of retinoid metabolic process genes *Adh1*, we cloned the promoter region of *ADH1* gene to drive the expression of luciferase. We found overexpression of *RBP-J* significantly decreased the relative *Adh1* promoter driven luciferase level, but co-transfection of both *RBP-J* and *N1ICD* increased the relative *Adh1* promoter driven luciferase level (*Figure 4F*). These results prove that the promoter region of *Adh1* is suppressed by

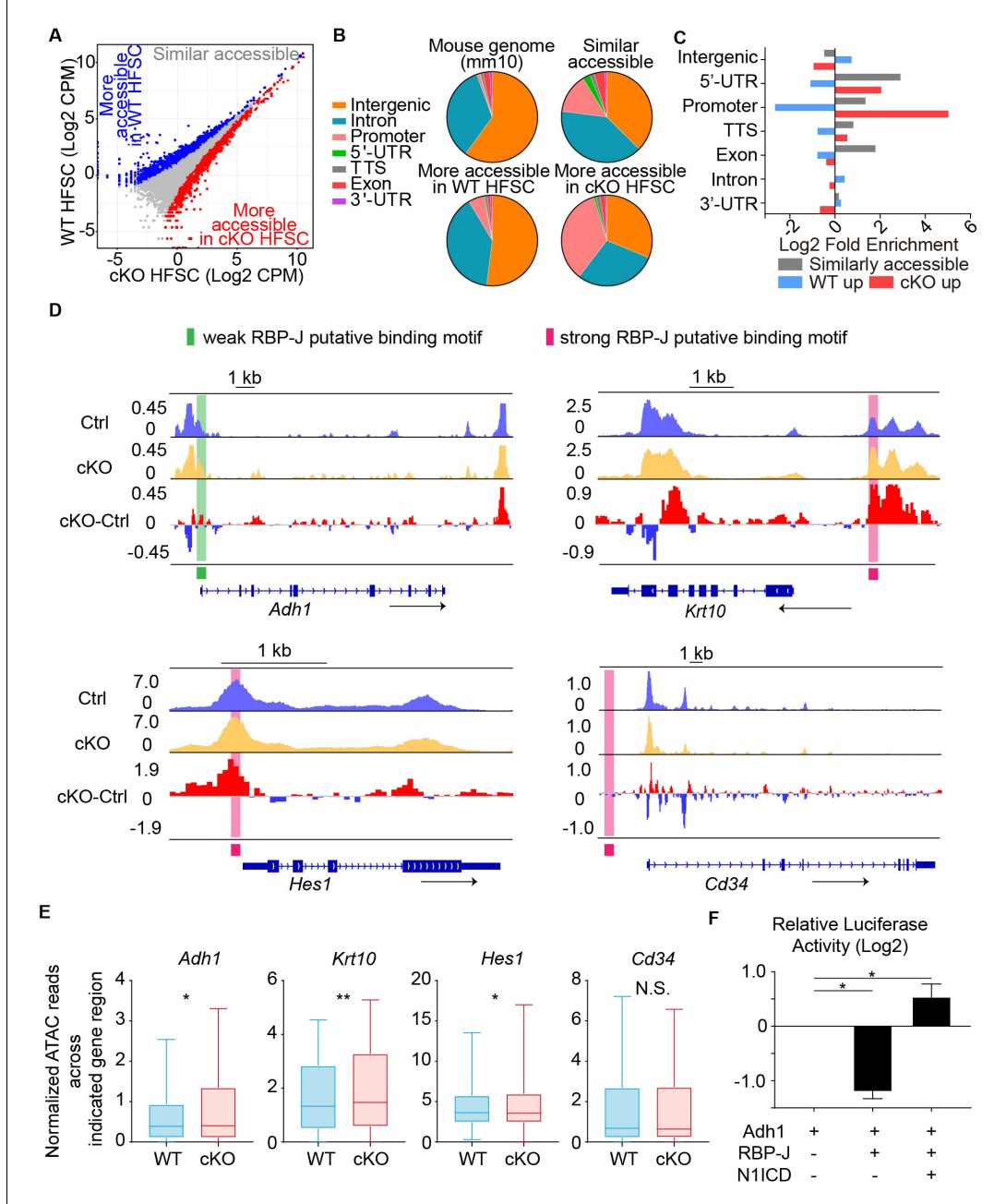

**Figure 4.** Chromatin accessibility change after loss of RBP-J in HFSCs. (A) ATAC-seq signal comparison between FACS isolated HFSCs from *WT* and *Krt6-CreER::RBP-J cKO* HFs at D2 after wax. Each dot represents an ATAC-seq peak. Regions with more accessibility in *cKO* than in *WT* cells (red) *vs.* more accessibility in *WT* than in *cKO* cells (blue) are indicated. (B) Pie chart showing the classification of ATAC-seq peaks from (A) compared with the mouse genome (mm10). (C) Log2-fold enrichment of the genomic classes shown in (B) relative to the whole genome. (D) ATAC-seq track signals at four representative loci (*Adh1, Krt10, Hes1* and *CD34*). RBP-J putative binding motifs are highlighted by pink (strong) and green (weak) boxes. (E) Statistic analysis of normalized ATAC-seq reads from 3.5 kb upstream to 3.5 kb downstream of indicated genes between *WT* and *cKO*. Data are showed in box-whisker from min to max. (*) p<0.05; (**) p<0.01. (F) In vitro luciferase assay for *Adh1* promoter activity with or without RBP-J or N1ICD using cultured mouse keratinocytes. N = 3 and data are mean ± SD. (*) p<0.05.

RBP-J, and the finding is consistent with the in vivo results. Taken together, after RBP-J ablation, the chromatin states of some RBP-J targeted genes were changed and became more open and resulted in higher transcriptional activity.

## Increased retinoic acid from RBP-J null HFSCs promotes ectopic differentiation of McSC in the niche by elevating c-Kit protein level of McSCs

To functionally test whether RA is involved in promoting McSC ectopic differentiation in the niche, we used the RA synthesis inhibitor WIN18446 to treat *WT* and *Krt6-CreER::RBP-J cKO* skin (*Figure 5A*). After treatment, RA level in *cKO* HFs was decreased from ~90 pg RA per million HFSCs to ~30 pg RA per million HFSCs, as measured by ELISA assay using FACS isolated HFSCs (*Figure 5B*). RA levels in other cell types (CPL, HG and matrix) are also examined and no significant difference was observed (*Figure 5—figure supplement 1A–C*). The ectopic pigmentation in *cKO* bulge was efficiently rescued by WIN18446 treatment, but not vehicle treatment (*Figure 5C*). Compared to ~80% of *cKO* HFs showing ectopic pigmentation in the bulge, only about 20% of *cKO* HFs did after treatment with WIN18446. Then we examined the mRNA level of melanocyte marker genes of FACS isolated McSCs from *WT* and *Krt6-CreER::RBP-J cKO* skin with WIN18446 or vehicle treatment (*Figure 5D*). We found that even though McSC markers such as *DCT* and *c-Kit* levels were similar in all four groups examined, melanocyte lineage master regulator MITF and melanocyte differentiation marker *Tyrp1* was significantly down regulated in McSCs isolated from *cKO* HFs after RA synthesis inhibitor WIN18446 treatment, consistent with the ectopic pigmentation change. This result revealed that RA is necessary to induce ectopic McSC differentiation in the aberrant niche of *Krt6-CreER::RBP-J cKO* HFs.

To test whether RA along is sufficient to trigger McSC differentiation in a *WT* niche at the telogen to anagen transition stage, we treated C57 *WT* dorsal skin with different amounts of RA and observed a dosage dependent effect (*Figure 5E*). At 2 nM ~200 nM RA dosage,~50% of dorsal skin HFs show ectopic pigmentation in the bulge at D4 after wax. Considering other cell types may also be affected by RA topical treatment in vivo, we FACS isolated primary telogen McSCs and cultured them in vitro and treated them with different concentration of RA (*Figure 5—figure supplement 1D–F*). We found that at 6 hr timepoint, 1 to 100 nM RA is sufficient to induce higher c-Kit and Tyrp1 level but at 12 hr timepoint, this induction is diminished (*Figure 5F*). This demonstrates RA can induce McSCs differentiation in a time and dosage dependent manner. Together these results indicate that elevated level of RA is both necessary and sufficient to induce ectopic McSC pigmentation in the niche in hair cycle dependent fashion.

During our FACS isolation of McSCs from telogen skin, we noticed a consistent pattern of increased c-Kit levels on McSCs isolated from the *Krt6-CreER::RBP-J cKO* and *Sox9-CreER::RBP-J cKO* mice compared to *WT*. Since c-Kit is the receptor of the KIT-ligand and plays an essential role in melanocyte maturation and differentiation (*Ito et al., 1999*), we used FACS to quantify the c-Kit protein level on McSCs (*Figure 5—figure supplement 1D–F*). We found that although percentage of total c-Kit+ McSC is not changed (*Figure 6—figure supplement 1A*), the percentage of c-Kit$^{high}$ cells among the c-Kit+ cells increased 2-fold in the *Krt6-CreER::RBP-J cKO* and *Sox9-CreER::RBP-J cKO* mice compared to *WT* (*Figure 6A*, *Figure 6—figure supplement 1B*). It has been reported that RA can increase c-Kit protein level by influencing its translation in testes tissue (*Busada et al., 2015*). Since RA synthesis inhibitor WIN18446 can efficiently rescue the ectopic pigmentation phenotype in *cKO*, we then asked whether or not the increased level of c-Kit on McSCs from *cKO* skin was due to the changed RA level in vivo. After WIN18446 treatment, the percentage of c-Kit$^{high}$/c-Kit+ did not change in *WT* skin, but the increased percentage of c-Kit$^{high}$/c-Kit+ from *Krt6-CreER::RBP-J cKO* skin was dramatically decreased to level slightly lower than *WT* (*Figure 6A*). The numbers of McSCs remain the same in *WT* and *cKO* telogen bulge after WIN18446 treatment (*Figure 6—figure supplement 1C*). Then we tested whether RA is sufficient to enhance c-Kit protein level. In vivo, we quantified c-Kit RFI(Relative Fluorescence Intensity) by immunofluorescence after topical RA treatment and found significant increase (*Figure 6B*). The effect of RA on c-Kit level in vivo was not due to change in transcription level because based on our qPCR result, the *c-Kit* mRNA levels were similar in FACS isolated McSCs from *WT* and *Krt6-CreER::RBP-J cKO* skin (*Figure 5D*). Mechanistically we found that when we over-expressed c-Kit 5'UTR-GFP fusion protein under the control ubiquitous promoter CMV in 293 t cells, GFP level is repressed if we co-expressed RA receptor RARα, and this repression can be released when 1 μM RA is added into the medium (*Figure 6C*) (*Poon and Chen, 2008*). To test whether or not the increased level of c-Kit on McSCs can lead to differentiation, we used in vitro assay to overexpress *c-Kit* in cultured primary melanocytes and analyzed melanocyte

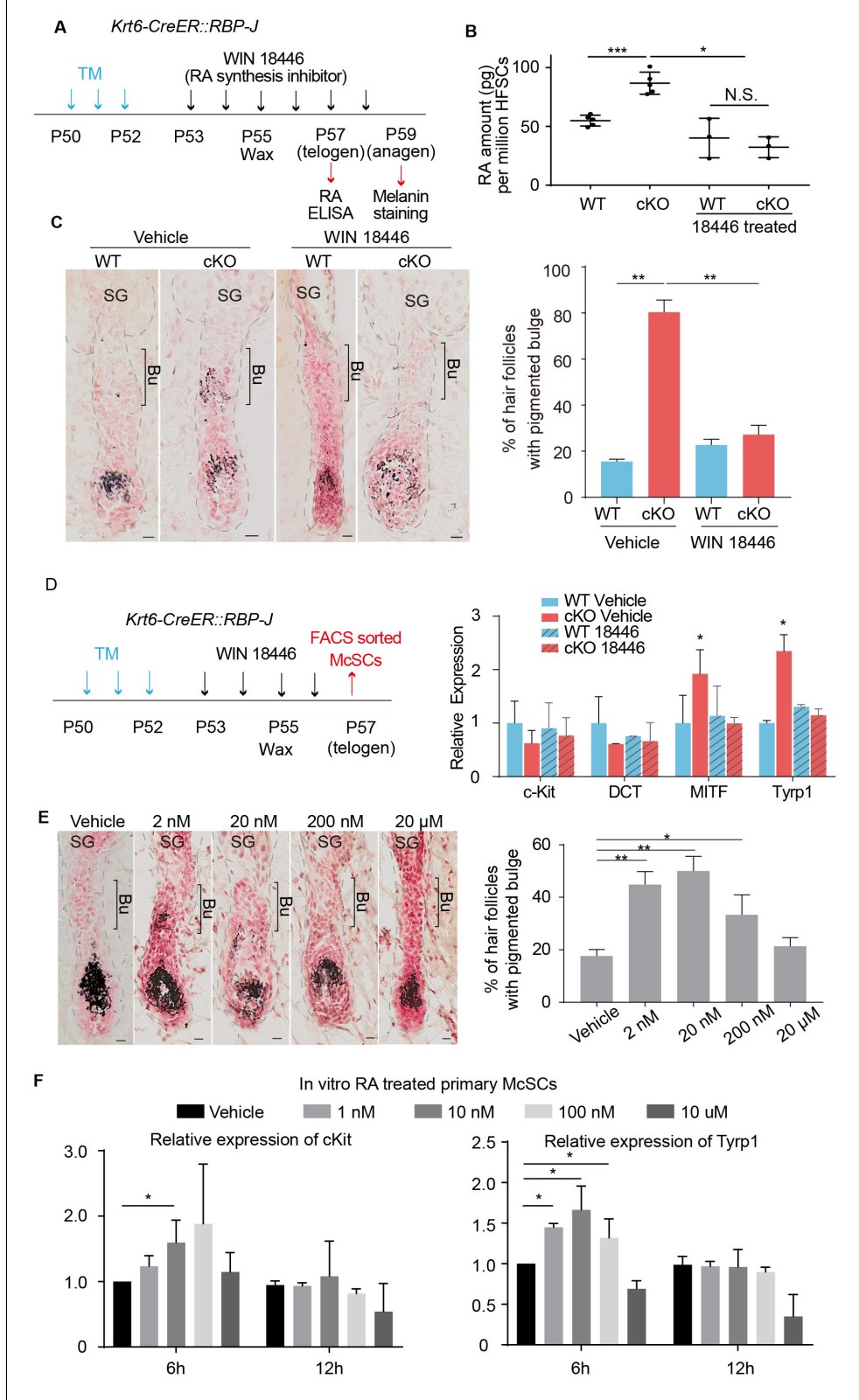

**Figure 5.** Increased RA from RBP-J null HFSCs causes McSCs to differentiate abnormally in the niche. (**A**) Schematic diagram of retinoic acid synthesis inhibitor treatment experiment. Tamoxifen was injected on P50-52 at telogen to induce ablation of *RBP-J* in *Krt6-CreER::RBP-J* mice, then CPL cells were removed by wax on P55 to induce anagen entry. Retinoic acid synthesis inhibitor WIN18446 or vehicle was administered from P53 to P58 spanning the telogen to anagen transition stage. HFSCs were harvested at P57 telogen for RA ELISA detection and dorsal skin samples were taken at P59 for

*Figure 5 continued on next page*

Figure 5 continued

analysis. (**B**) ELISA assay to measure the level of retinoic acid in FACS isolated HFSCs from *WT* and *Krt6-CreER::RBP-J cKO* dorsal skin after different treatments. (**C**) Representative dorsal skin Masson-Fontana staining images and quantification of ectopic pigmentation in the bulge of *WT* and *Krt6-CreER::RBP-J cKO* HFs after different treatments as indicated in (**A**). (**D**) Schematic diagram and qPCR analysis of melanocyte lineage or differentiation related genes in FACS isolated McSCs from *WT* and *Krt6-CreER::RBP-J cKO* HFs after indicated treatments. (**E**) Representative dorsal skin Masson-Fontana staining images and quantification of ectopic pigmentation in the bulge of *WT* HFs after topical treatment with indicated amount of RA treatment. CPL cells were removed by wax on P55 to induce anagen entry. Retinoic acid was administered from P53 to P58 spanning the telogen to anagen transition stage in *WT* mice. Dorsal skin samples were taken at P59 for analysis. (**F**) QPCR analysis of c-Kit and Tyrp1 expression level on cultured primary McSCs isolated from telogen HFs after treatment of RA with indicated dosage and time. All data are expressed as mean ± SD >= 15 follicles are quantified each mouse. N >= 3. (*) p<0.05; (**) p<0.01; (***) p<0.001. Scale bars = 10 μm.

The online version of this article includes the following figure supplement(s) for figure 5:

**Figure supplement 1.** FACS sorting strategy, validation and quantification of McSCs and other cell types.

differentiation marker *Tyrp1* by qPCR (*Figure 6D*). The culture medium of melanocytes mandates addition of the KIT-ligand SCF. In the presence of ligand, overexpression of *c-Kit* in melanocytes resulted in about two fold increase in the *Tyrp1* mRNA level, which is similar to the level of increase observed in the FACS isolated McSCs from *cKO* skin. Taken together, these results show that increased level of RA from the RBP-J null HFSCs leads to increased level of c-Kit protein on McSCs and the ectopic pigmentation phenotype in the mutant niche.

## Hair cycle dependent increase of HFSC secreted SCF induces aberrant differentiation of RA sensitized McSC in the shared niche

To directly test whether the increased level of c-Kit on McSCs is responsible for the ectopic differentiation phenotype in vivo, we blocked the downstream signaling pathways of c-Kit using inhibitors. There are two major pathways that function downstream of c-Kit: the MEK pathway and the mTOR pathway, which can be efficiently blocked by PD0325901 and rapamycin respectively (*Barrett et al., 2008*; *Edwards and Wandless, 2007*). The mTOR pathway inhibitor rapamycin can effectively arrest HFs in telogen as reported before (*Kellenberger and Tauchi, 2013*). But the ectopic pigmentation phenotype still persisted in *Krt6-CreER::RBP-J cKO* bulge after rapamycin treatment (*Figure 7—figure supplement 1A*). On the other hand, we observed higher MEK downstream pERK signaling in McSCs in cKO follicles (*Figure 7A*), and also found treatment of MEK pathway inhibitor PD0325901 efficiently rescued the ectopic pigmentation phenotype in mutant skin (*Figure 7B*). The number of McSCs remains the same in *WT* and *cKO* telogen bulge after PD0325901 treatment (*Figure 7—figure supplement 1B*). Notably, the PD0325901 treat can also significantly diminish the differentiation of melanocyte in the matrix of anagen HFs, which is consistent with the known function of c-Kit for melanocyte terminal differentiation (*Ito et al., 1999*). So, together these results confirmed that the increased level of c-Kit on McSCs is required for the ectopic differentiation of McSCs via the c-Kit-MEK pathway.

The SC niche is often considered to be devoid of differentiation signals that might impair the self-renewal of SCs. So we were curious about where the source of the KIT-ligand would be for the McSC c-Kit receptor to get activated and initiate the ectopic differentiation program. The hair cycle dependent ectopic differentiation of McSCs peaks at D3 post anagen initiation. At this stage the HFs are elongated with the matrix and dermal papilla far away from the bulge region where the ectopic pigmentation occurs. Considering the distance between the matrix and bulge at this time point, we turned our focus to the closest neighbor of McSCs: the HFSCs. First we did qPCR analysis of KIT-ligand *SCF* in FACS isolated HFSCs from *WT* and *Krt6-CreER::RBP-J cKO* dorsal skin at both telogen and anagen. At both hair cycle stages we did detect *SCF* expression in the HFSCs but there was no difference between the *WT* and *cKO*. However, when HFSCs transitioned from telogen to anagen, there were significant increase in the level of *SCF* mRNA in both *WT* and *cKO* (*Figure 7C*).

To functionally test whether or not the observed increase of KIT-ligand *SCF* in HFSCs at the telogen to anagen transition stage is the source of differentiation signal that triggers the hair cycle dependent ectopic differentiation of McSCs in the mutant niche, we genetically ablated *SCF* together with *RBP-J* to see if this can rescue the phenotype. When compared to *Krt6-CreER::RBP-J cKO*, the *Krt6-CreER::RBP-J::SCF cKO* completely rescued the McSC ectopic differentiation phenotype in the bulge area without affecting terminal differentiation of melanocytes in the matrix

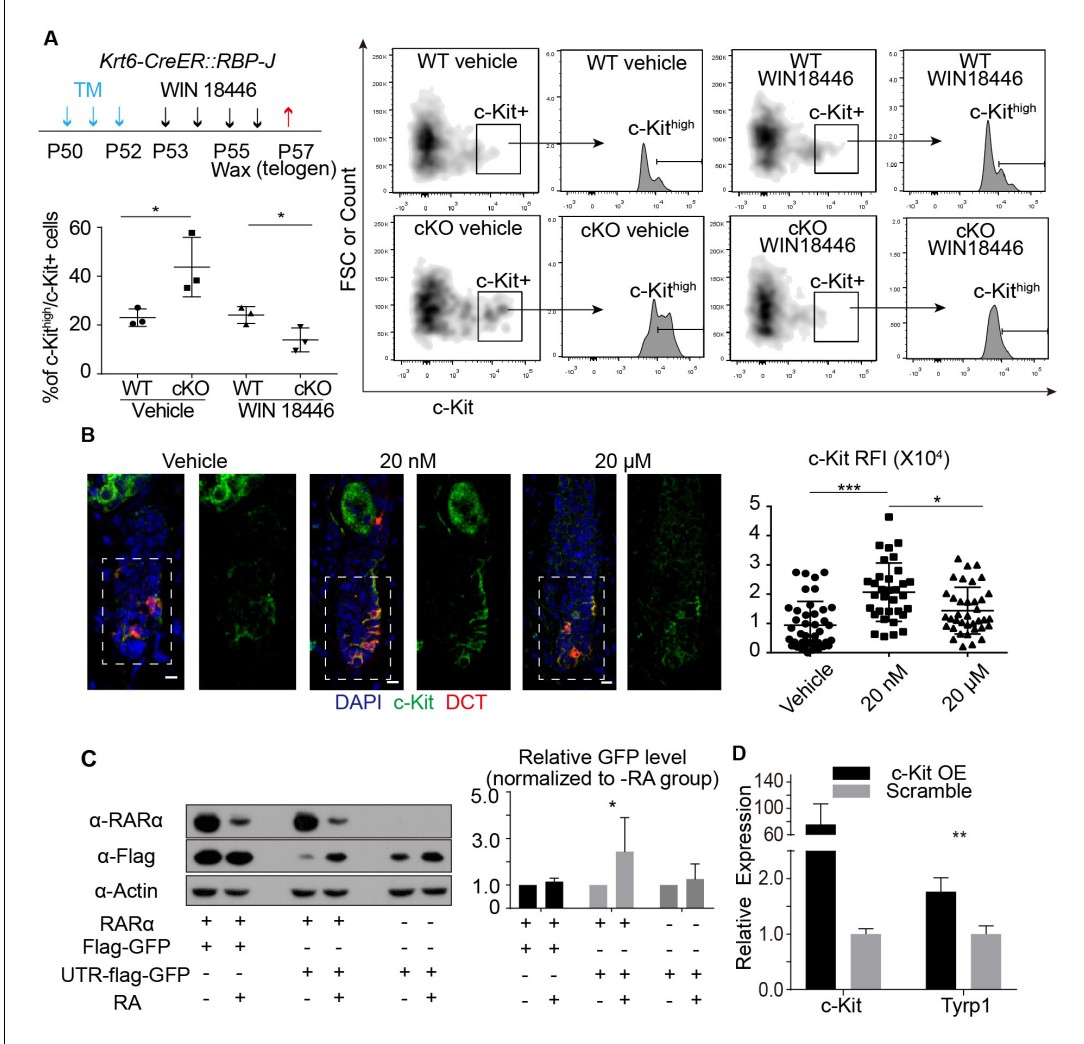

**Figure 6.** RA from cKO HFSCs enhances c-Kit protein level on McSCs that leads to abnormal differentiation. (**A**) Schematic diagram, FACS profiles and quantification of c-Kit+ McSCs in telogen HFs from *WT* and *Krt6-CreER::RBP-J cKO* mice after indicated treatments. Tamoxifen was injected on P50-52 at telogen to induce ablation of *RBP-J* in *Krt6-CreER::RBP-J* mice, then CPL cells were removed by wax on P55 to induce anagen entry. Retinoic acid synthesis inhibitor WIN18446 or vehicle was administered from P53 to P56. Dorsal skin samples were taken at P57 at telogen for analysis. c-Kithigh population was further gated from c-Kit+ population and quantified by c-Kithigh/c-Kit+ percentage. (**B**) Representative immunofluorescence images and quantification of c-Kit RFI (Relative Fluorescence Intensity) after indicated topical RA treatments on C57 WT mice. (**C**) Western blot and quantification of GFP or UTR-GFP protein level with or without RARα and RA. (**D**) QPCR analysis of Tyrp1 mRNA level in cultured primary melanocytes with or without overexpression of c-Kit. All data are expressed as mean ± SD ≥ 15 follicles are quantified each mouse. N = 3. (*) p<0.05; (**) p<0.01; (***) p<0.001. The online version of this article includes the following figure supplement(s) for figure 6:

**Figure supplement 1.** Quantification of c-Kit level on McSCs and numbers of melanocytes.

(*Figure 7D*). Furthermore, the expression of melanocyte differentiation marker Tyrp1 was also down regulated in the *Krt6-CreER::RBP-J::SCF cKO* HFs (*Figure 7E*). This genetic rescue experiment confirms that it is the KIT-ligand originated from HFSCs that activates the c-Kit-MEK pathway to induce McSC ectopic differentiation in the mutant niche. We also checked SCF level in rapamycin treated mice and found although hair cycle is delayed after rapamycin treatment, SCF level is still increased in HFSCs comparing to that in telogen (*Figure 7—figure supplement 1C*), which can explain the occurrence of ectopic pigmentation. It's worth emphasize that the level of KIT-ligand *SCF* in HFSCs was not changed between *WT* and *Krt6-CreER::RBP-J cKO* skin (*Figure 7C*). So this striking rescue effect by ablation of *SCF* revealed that the McSCs niche environment is not devoid of differentiation signals. McSC maintain self-renewal ability by expressing low level of c-Kit to remain insensitive to

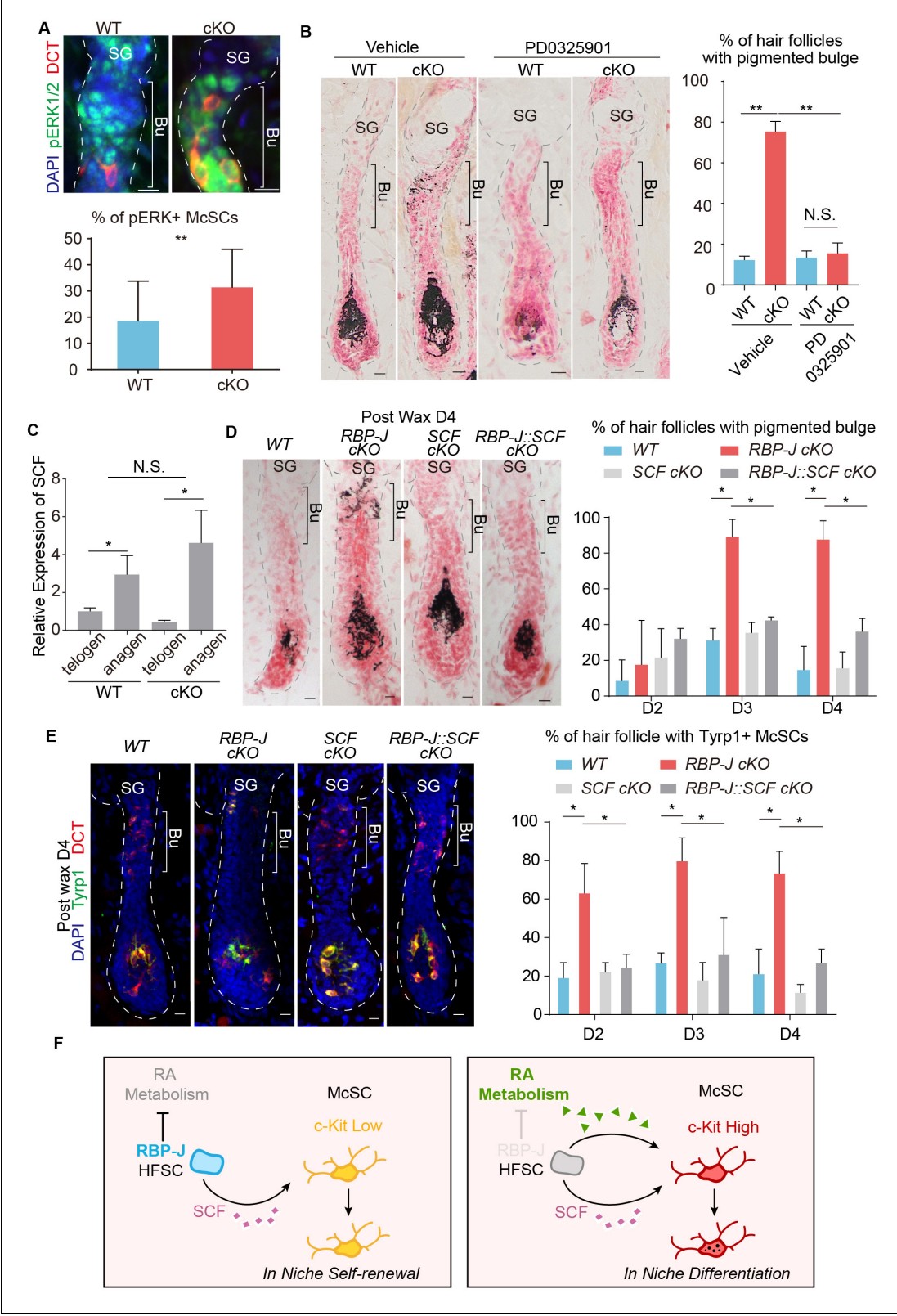

**Figure 7.** Hair cycle dependent increase of HFSC secreted SCF induces aberrant differentiation of RA sensitized McSC in the shared niche. (**A**) Representative immunofluorescence images and quantification of pERK+ McSCs in *WT* and *Krt6-CreER::RBP-J cKO* HFs. (**B**) Representative dorsal skin Masson-Fontana staining images and quantification of ectopic pigmentation in the bulge of *WT* and *Krt6-CreER::RBP-J cKO* HFs after treatment with vehicle or MEK inhibitor PD0325901. (**C**) QPCR analysis of *SCF* mRNA in FACS isolated HFSCs from *WT* and *Krt6-CreER::RBP-J cKO* dorsal skin at both
*Figure 7 continued on next page*

*Figure 7 continued*

telogen and anagen. (**D**) Representative dorsal skin Masson-Fontana staining images and quantification of ectopic pigmentation in the bulge of *WT, Krt6-CreER::RBP-J, Krt6-CreER::SCF and Krt6-CreER::RBP-J::SCF cKO* HFs at the telogen to anagen transition stage. Tamoxifen was injected on P50-52 at telogen, then CPL cells were removed by wax on P54 to induce anagen entry. Dorsal skin samples were taken at D4 after wax. HFs were counter-stained by neutral red. Brackets indicate the bulge region where McSC ectopic differentiation is observed. (**E**) Representative immunofluorescence images and quantification of percentage of HF with Tyrp1+ McSCs. McSC located in the bulge area indicated by brackets were labeled with DCT staining. Melanocyte differentiation marker Tyrp1 in DCT+ McSCs located in the bulge indicates ectopic differentiation. Percentage of HFs with Tyrp1+ McSCs in *WT, Krt6-CreER::RBP-J, Krt6-CreER::SCF and Krt6-CreER::RBP-J::SCF cKO* HFs in dorsal skin were quantified. (**F**) Schematic diagram of HFSC expressed RBP-J regulates retinoic acid metabolic process to maintain the differentiation refractory niche for neighboring McSC. During telogen to anagen transition the SCF ligand emanating from HFSCs increases, but co-occupant McSCs in the shared niche do not respond to the differentiation signal due to their low c-Kit protein level. In the absence of *RBP-J* in HFSCs, retinoic acid is increased after de-repression of retinoid metabolic process genes such as *Adh1*. Increased retinoic acid from HFSCs enhances c-Kit protein level in McSC and sensitizes them to the increased SCF level in bulge at the telogen to anagen transition stage. This results in defect in McSC self-renewal and ectopic differentiation in the niche. All data are expressed as mean ± SD ≥ 15 follicles are quantified each mouse. N = 3. (*) p<0.05; (**) p<0.01.

The online version of this article includes the following figure supplement(s) for figure 7:

**Figure supplement 1.** Effect of Rapamycin or RA treatments on McSCs.

the differentiation signals. While their neighboring HFSCs are not just passive receiver of signaling inputs from the niche, they actively suppress differentiation-sensitizing signals in the niche, such as RA level, to help maintain a differentiation refractory niche environment (*Figure 7F*).

## Discussion

Fate determination of normal and malignant epidermal cells can be influenced by RA level in culture (*Kopan et al., 1987*; *Kopan and Fuchs, 1989*), but it is not known whether endogenous RA can regulate skin SC function in vivo. Here we uncovered an unexpected role of HFSC regulating the RA level in vivo by RBP-J suppressed retinoid metabolic process genes. The increased endogenous level of RA emanating from HFSCs as a result of RBP-J loss disrupts the self-renewal of McSCs in the same niche space. The fact a metabolite, RA, is the messenger mediating the cross talk of two types of SCs in a shared niche, revealed the previously unknown function of metabolite as a signaling molecule in vivo. Next it would be essential to probe more broadly into whether physiological variations of other metabolite levels could impact on SC fate choices and function during tissue regeneration and disease development, as indicated by recent finding in hematopoietic system (*Agathocleous et al., 2017*). Previous studies have revealed that Nfib/Edn signaling can regulate the differentiation of McSCs in collaboration with KIT or Wnt signaling (*Chang et al., 2013*; *Takeo et al., 2016*). In our RNA-seq, Nfib expression is the same in Ctrl and cKO HFSCs (232.198 v.s. 203.073). In the Nfib mutants mice the melanocyte number is increased in bulge that is not observed in our RBP-J mutants. Edn family genes are either not expressed or shows decreased expression in cKO HFSCs. (Edn1: 0.237 v.s. 0.155; Edn2: 4.10 v.s. 1.75; Edn3: 0.073 v.s. 0). We have also functionally ruled out Wnt playing a role in the phenotype. Besides the well-studied morphogens, we discover that metabolites represent an entirely new realm of signaling molecules that mediates SC and niche cross talk, and their potential functions warrant further studies in the other systems under homeostasis and pathological conditions.

Given the current understanding that niche plays dominant roles in SC fate determination, maintenance, and even disease initiation (*Xu et al., 2015*; *Scadden, 2007*; *Clevers et al., 2014*), SCs are often perceived as passive recipients of the niche signaling and act as they were told. Recent reports started to emerge that in some cases SCs actually play instructive roles in the physical niche formation during morphogenesis (*Tamplin et al., 2015*; *Seleit et al., 2017*). Whether these are exceptions rather than the rule, and whether after the niche has been established, SCs can also regulate the niche signaling environment are less clear. Unexpectedly, we found here that the niche shared by HFSCS and McSCs is not devoid of differentiation signal. In fact the HFSCs secreted KIT-ligand is sufficient to induce the differentiation of neighboring McSCs in the niche, when the c-Kit protein level on McSCs is elevated by increased RA levels in the niche. During normal homeostasis, the Notch pathway co-factor RBP-J functions in HFSCs to suppress the retinoid metabolic process

genes. This reveals that HFSCs play an instructive role in maintaining a differentiation refractory niche environment for the neighboring McSCs.

Retinoid is a common ingredient in daily makeup or cream for acne treatment. If RA can induce McSC differentiation and possible stem cell loss later on, it will become a concern. To test if McSCs will lose their number, we treated C57 WT mice with topical RA and followed McSC number after one hair cycle (*Figure 7—figure supplement 1D*). We found that under concentration (1 μg) similar to daily use (1 ~ 5 μg), there is no obvious difference of McSC number comparing to vehicle group, which means RA is safe and will not lead to McSC loss during our daily care. Interestingly, when we treated with lower concentration (1 ng), there is a slight but significant increase in McSC number. This may result from some compensational mechanism between stem cells and possibly could be used to restore melanocyte number in natural hair graying process and diseases that result in melanocyte loss.

## Materials and methods

### Mice and animal treatment

*Sox9-CreER* (*Xu et al., 2015*) and *Krt6-CreER* were generated by the NIBS transgenic center. We generated the *Krt6-CreER* mice by integrating IRES-CreERT2-SV40pA cassettes into the 3' UTR of the endogenous mouse *Krt6a* gene via Cas9/RNA mediated gene targeting in zygotes. The insertion site is before the tgctcagagccccgagttcaggg sequence. *RBP-J flox* mice were kindly provided by Dr. Tasuku Honjo (*Oka et al., 1995*). Nfatc1-CreER mice were generated and provided by Dr. Bing Zhou (*Tian et al., 2017*). K14-H2BGFP mice were kindly provided by Dr. Elaine Fuchs. *SCF flox* mice have been described previously (*Ding et al., 2012*). cKO was achieved with intraperitoneal injection with 10 mg/mL tamoxifen (Sigma) in sunflower oil (Sigma) (1 mg per 10 g body weight every time) from P1 to P4 during morphogenesis or from P50 to P52 during homeostasis. Wax was performed at P54 without specification. For Wnt inhibitor treatment, LGK974 was formulated 5 mg/mL in 0.5% MC/0.5% Tween 80 and administered by oral gavage at 3 mg/kg body weight every time from P54 to P57. Dorsal skin was harvested at P58. For RA synthesis inhibitor treatment, WIN 18446 was suspended in 1% gum tragacanth and administered orally 100 μg/g body weight each time from P53 to P58. Dorsal skin was harvested at P59. For mTOR inhibitor treatment, 10 mg/ml rapamycin in ethanol was diluted in a solution of 5% Tween 80 and 5% PEG-400 in sterile water and then injected intraperitoneally 0.5 μg/g body weight each time from P53 to P58. Dorsal skin was harvested at P59. For MEK inhibitor treatment, PD0325901 was formulated in a solution of 30% PEG 400 and 5% Tween 80 in sterile water and administered orally 20 mg/kg body weight from P54 to P57. Dorsal skin was harvested at P58. All mice were maintained in an SPF facility, and the procedures used were consistent with the National Institute of Biological Sciences guide for the care and use of laboratory animals.

### Histology and immunofluorescence

Whole-mount tails were treated with 50 μM EDTA to separate the epidermis from the dermis. Melanin staining was performed using a Fontana–Masson stain kit (Junrui biotech Inc) according to the product instructions. For frozen sections, tissues were embedded in O.C.T compound and frozen. Sections (15–30 μm) were fixed in 4% paraformaldehyde (PFA) for 10 min, washed, and blocked for 1 hr in blocking buffer (2% normal donkey serum, 1% BSA, 0.3% Triton in PBS). MOM Basic kit (Vector Laboratories) was used for blocking when primary antibodies were generated from mouse. Primary antibodies were diluted in blocking buffer and incubated at 4°C O/N. Samples were washed in PBS and incubated with 2nd antibody for 1 hr at RT. Samples were then washed and mounted with DAPI in 50% glycerol for imaging. Immunofluorescence images were taken with Nikon A1-R confocal microscope and analyzed with ImageJ. N1ICD, RBP-J, and β-catenin were amplified with ABC kit (vector) according to the manufacturer's protocols. The antibodies used were as follows: anti-DCT (homemade), anti-CD34 (ebioscience), anti-RBP-J (active motif), anti-Tyrp1 (homemade), anti-β-catenin (Sigma), anti-Krt14 (homemade), anti-Sox9 (homemade), anti-Krt6 (homemade), anti-cleaved Notch1(CST), anti-P-cad (R and D), anti-cleaved caspase 3 (CST), anti-yH2Ax (CST) and anti-pERK1/2 (CST).

## Isolation of cells and FACS

For isolation of anagen cells, skins were first incubated in 0.25% collagenase (Sigma) in HBSS (GIBCO) at 37°C for 1–2 hr to digest the dermis. Next, the digested tissues were centrifuged and resuspended in 0.25% Trypsin (GIBCO) and incubated at 37°C for 20 min with gentle shaking. Single-cell suspensions were obtained by pipetting gently and filtering through 70 μm strainers, followed by 40 μm strainers. Staining buffer (PBS with 5% FBS treated with Bio-Rad Chelex to remove calcium) was added to inactivate trypsin, and cells were collected by centrifugation for 5 min at 300 × g. Cell suspensions were incubated with the appropriate antibodies diluted in staining buffer for 15 min at 4°C. For the isolation of telogen cells, subcutaneous fat was removed from skins with a scalpel. Skins were placed on 0.25% trypsin at 37°C for 30 min with the dermis side facing down. Skins were then gently scalped from the epidermis to isolate epidermal cells and filtered, centrifuged, and stained as above. The following antibodies were used: CD34–Alexa 660 (eBioscience), α6–phycoerythrin (eBioscience), Sca1–PE-Cy7 (eBioscience), and c-Kit-APC (eBioscience).

## Western blot analysis

Denatured protein extracted from isolated cells was loaded on a 10% SDS-PAGE gel, electrophoresed, and then transferred onto a PVDF membrane. PVDF membranes were blocked with 5% fat-free milk, and the membranes were then probed with anti-N1CD (CST), anti-N2ICD (CST), anti-Actin-HRP (MBL), anti-Flag (Abmart), anti-RARα (Santa Cruz) and anti-mouse-p-MLKL (Abcam) at 4°C O/N. Blots were then incubated with secondary antibody. Peroxidase activity on the membrane was visualized on X-ray film using the ECL western blotting detection system.

## RNA isolation and quantitative PCR

Total RNA was isolated from FACS-purified cells lysed with Trizol (GIBCO) followed by extraction using a direct-Zol RNA mini prep kit (Zymo research). An equal amount of RNA was added to a reverse-transcriptase reaction mix (Vazyme). Expression levels were normalized to PPIB. Quantitative PCR was conducted using a CFX96TM Real-Time system (Bio-RAD) with SYBR FAST qPCR Master Mix (2X) (Kapa). All primer pairs were designed for the same cycling conditions, which were: 3 min at 95°C; 40 cycles of 5 s at 95°C, 30 s at 60°C. qPCR primers are listed in *Supplementary file 2*.

## ELISA

HFSCs and other cell types are isolated as mentioned above and 0.5 million cells are harvested and washed three times with ice cold PBS to remove serum. Cells are resuspended in 100 μL RIPA buffer (50 mM Tris pH8.0, 150 mM NaCl, 1.0% NP40, 0.5% DOC, 0.1% SDS) and lyse with interval vortex on ice for 30 min. After centrifugation, supernatant is transferred into a new tube to detect RA with Mouse Retinoic Acid ELISA kit (Cusabio). ELISA follows instructions on the website of Cusabio (www.cusabio.com). Minimal light exposure is ensured during whole process.

## In vitro luciferase reporter assay

Adh1 promoter sequence (5 kb upstream of TSS site) is cloned into luciferase reporter vector. Keratinocytes are transfected with indicated plasmids by lipofectamine 3000. 48 hr later transfected cells are lysed to detect luminescence with Bright-Glo luciferase assay system from Promega.

## In vitro RARα and c-Kit 5'UTR interaction assay

293 t cells are transfected with pcDNA-RARα and 24 hr later transfected with pcDNA-GFP or pcDNA-UTR-GFP. 3 hr post GFP transfection, vehicle or 1 μM RA is added for 3 hr and cells are harvested and lysed for western blot.

## ATAC-seq

ATAC-seq is performed with freshly FACS isolated HFSCs according to Omni-ATAC protocol (*Corces et al., 2017*). Briefly, 50,000 live cells are harvested and resuspended in 50 μL ATAC-Resuspension Buffer (RSB) containing 0.1% NP40, 0.1% Tween-20, and 0.01% Digitonin and incubate on ice for 3 min. The lysis is washed out with 1 mL cold ATAC-RSB containing 0.1% Tween-20 and invert tube three times to mix. Spin to pellet nuclei and resuspend in 50 μL of transposition mixture by pipetting up and down six times. Transposition mix = (25 μL 2x TD buffer, 2.5 μL transposase (100

nM final), 16.5 µL PBS, 0.5 µL 1% digitonin, 0.5 µL 10% Tween-20, 5 µL H2O). Incubate reaction at 37°C for 30 min in a thermomixer with 1000 RPM mixing. The transposed DNA was recovered with Zymo DNA clean and concentrator-5. Library was prepared with Nextera DNA library Prep Kit. PE-150 was chosen as sequencing method.

## Sequencing data analysis

RNA samples from FACS-purified cells were submitted to the Biodynamic Optical Imaging Center of Peking University for quantification, RNA-seq library preparation, and sequencing. The libraries were sequenced on the Illumina HiSeq 2500 platform using the Pair-End 2 × 100 bp sequencing strategy. For analysis, data were mapped to the mouse genome (GRCm38/mm10), using TopHat (v2.0.13) with the default settings. Genes with significantly differential expression ($p \leq 0.05$ and FC > 1.5) were chosen for further analysis. Gene ontology (GO) analysis of genes was done using DAVID (Database for Annotation, Visualization, and Integrated Discovery). For gene set enrichment analysis (GSEA, Broad Institute), FPKM values (RNA-seq) were compared to the curated gene sets (C2), and the gene ontology gene sets (C5) of the Molecular Signature Database (MsigDB) using the Signal2Noise metric and 1000 gene set-based permutations. ATAC-seq reads were aligned to the mouse genome (mm10) using Bowtie2 (*Langmead and Salzberg, 2012*). Mitochondrial reads were excluded from downstream analysis. Data tracks were visualized by Integrative Genomics Viewer (IGV). Peak calling was performed by MACS2 (*Zhang et al., 2008*). ChIP-seq data were obtained from previous paper (*Castel et al., 2013*; *Meredith et al., 2013*). RBP-J binding motif was discovered with Homer2 de novo motif analysis. Core binding motif discovered was GTG(G/A)GAA (strong), GCG(G/A)GAA (weak) according to previous paper (*Bartels et al., 2011*).

## Acknowledgements

This work was supported by grants from the National Key R and D Program of China (2017YFA0103500) and the National Basic Research Program of China 973 Programs (2012CB518700 and 2014CB849602). We are grateful to the NIBS Animal Facility for their expert handling and care of mice, the NIBS Transgenic Animal Center for generation of gene-editing mice, the NIBS Biological Resource Centre for FACS sorting, the NIBS imaging facility for assistance with the microscope experiment. We thank all members of the Chen lab for discussions and technical support. We especially thank Dr. Haibing Wang from the Institute of Zoology at Chinese Academy of Science for helping us import the RBP-J flox mice.

## Additional information

### Funding

| Funder | Grant reference number | Author |
| --- | --- | --- |
| National Key R&D Program of China | 2017YFA0103500 | Ting Chen |
| National Basic Research Program of China (973 Program) | 2014CB849602 | Ting Chen |
| National Basic Research Program of China (973 Program) | 2012CB518700 | Ting Chen |

The funders had no role in study design, data collection and interpretation, or the decision to submit the work for publication.

### Author contributions

Zhiwei Lu, Conceptualization, Data curation, Formal analysis, Conceived the project, Designed the experiments, Wrote the manuscript, Performed majority of the experiments; Yuhua Xie, Data curation, Investigation, Participated in some of the in vivo experiments; Huanwei Huang, Data curation, Software, Analyzed the sequencing data; Kaiju Jiang, Data curation, Formal analysis, Performed some of the immunofluorescent staining experiments; Bin Zhou, Resources, Generated and provided the Nfatc1-CreER mice; Fengchao Wang, Resources, Generated CRISPR/Cas9-mediated gene-

editing mice; Ting Chen, Conceptualization, Resources, Supervision, Funding acquisition, Investigation, Conceived the project, Designed the experiments, Wrote the manuscript

### Author ORCIDs
Zhiwei Lu (iD) https://orcid.org/0000-0003-1777-4611
Yuhua Xie (iD) https://orcid.org/0000-0002-2014-7320
Ting Chen (iD) https://orcid.org/0000-0002-7404-4538

### Ethics

Animal experimentation: This study was performed in strict accordance with the recommendations in the Guide for the Care and Use of Laboratory Animals of the National Institutes of Biological Sciences. All of the animals were handled according to the guidelines of the Chinese law regulating the usage of experimental animals and the protocols (M0020) approved by the Committee on the Ethics of Animal Experiments of the National Institute of Biological Sciences, Beijing.

### Decision letter and Author response
Decision letter https://doi.org/10.7554/eLife.52712.sa1
Author response https://doi.org/10.7554/eLife.52712.sa2

## Additional files

### Supplementary files
• Supplementary file 1. Genes Up-regulated in RBP-J cKO versus WT dorsal HFSCs. RNA-seq analysis of *WT* and *cKO* HFSCs isolated from *Krt6-CreER::RBP-J* mice at D2 after wax to profile their transcriptome difference. Since RBP-J serves as a repressor in HFSCs, genes regulated >1.5 fold (p<0.05) in *cKO* compared to *WT* HFSCs were mainly focused.

• Supplementary file 2. Primer sequences used. Primer sequences related to *Figures 2*, *3*, *5*, *6* and *7* and *Figure 5—figure supplement 1*

• Transparent reporting form

### Data availability

All data generated or analysed during this study are included in the manuscript and supporting files.

The following dataset was generated:

| Author(s) | Year | Dataset title | Dataset URL | Database and Identifier |
|---|---|---|---|---|
| Zhiwei Lu, Yuhua Xie, Huanwei Huang, Kaiju Jiang, Ting Chen | 2020 | HFSC seq | http://bigd.big.ac.cn/bioproject/browse/PRJCA002151 | National Genomics Data Center, PRJCA002151 |

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
