## [Decision Letter]

**Acceptance summary:**

This manuscript demonstrates a very interesting role of HFSCs in maintaining melanocyte stem cells in their undifferentiated state through regulating retinoid metabolism. They link Notch signaling through the transcriptional mediator Rpbj to repression of a key gene for the retinoid metabolic pathway Adh1. Overall, this is a high quality and important work. Using several mouse models together with genomics, gene regulatory assays, and pharmacological experiments, the authors showed how HFSCs keep McSCs in their undifferentiated state via a metabolite. The analyses are rigorous and thorough, and the story will be of great interest to a broad readership of *eLife*, especially those in stem cell and developmental biology.

**Decision letter after peer review:**

Thank you for submitting your article "Hair follicle stem cells maintain anti-differentiation niche for melanocyte stem cells by regulating retinoid metabolism" for consideration by *eLife*. Your article has been reviewed by two peer reviewers, one of whom is a member of our Board of Reviewing Editors, and the evaluation has been overseen by Marianne Bronner as the Senior Editor. The following individual involved in review of your submission has agreed to reveal their identity: Ya-Chieh Hsu (Reviewer #2).

The reviewers have discussed the reviews with one another and the Reviewing Editor has drafted this decision to help you prepare a revised submission.

Summary:

This manuscript demonstrates a very interesting role of HFSCs in maintaining melanocyte stem cells in their undifferentiated state through regulating retinoid metabolism. They showed that Rpbj, a downstream component of the Notch pathway, directly represses a key gene for the retinoid metabolic pathway Adh1. Depletion of RBPj in bulge stem cells leads to increase levels of RA, which they showed to be functionally important to induce McSC differentiation in the niche by elevating the c-kit receptor on McSCs. Interestingly, they found that the SCF is expressed by the bulge stem cells and is elevated during telogen to anagen transition, which might account for the specific ectopic differentiation timing seen in RBPj mutants (telogen to anagen transition). Confirming this model, double knockout of RPBJ and SCF in bulge stem cells suppresses McSCs differentiation. Overall, this is a high quality and important work.

Essential revisions:

The reviewers' thought your manuscript needed a few points clarified that can likely be handled with textual changes.

1) Clarification of how quantifications were performed. Could the authors clarify how Tyrp1+ cells were quantified in their analyses in Figure 2.

2) The analysis of cell proliferation should be improved. The authors analyze Ki67+ cells but melanocyte stem cells may not express Ki67. The results could be explained by proliferation of melanocyte stem cells and by increasing numbers, more cells differentiate.

3) Does RA increase cKit expression or the number of ckit high cells? This part of the manuscript is not clear to me and seems essential for the proposed model.

4) HFSCs are known to regulate McSCs via Wnt and Edn signaling. In addition, loss of Nfib, a transcription factor in HFSCs, showed a similar phenotype as the RBPj mutant described here. What are the similarities and differences between the RBPj mutant and some of these mutants that have been described before (in addition to RA being a metabolite)? The authors' story is a substantial departure from these previous papers regarding cellular and molecular mechanisms. This said, discussing some of these previous findings in greater depth will help the readers to appreciate the authors' findings in the right context and also help to differentiate the authors' findings from the previous ones even better. It might also be helpful to look into the authors' existing RNAseq data to see if Nfib/Edn signaling levels might be different in Rbpj mutant.

5) The manuscript can benefit from some trimming. The detailed descriptions of the data generated made the current manuscript quite complex and less accessible to readers. This is a really excellent work with strong data support for all the claims. Focusing on elaborating the main messages while going light on the description of supportive /non-essential data will help the story to stand out even more and will help the readers from diverse fields to appreciate the beauty of the story.

---

## [Author Response]

Essential revisions:The reviewers' thought your manuscript needed a few points clarified that can likely be handled with textual changes.1) Clarification of how quantifications were performed. Could the authors clarify how Tyrp1+ cells were quantified in their analyses in Figure 2.

In Figure 2J, we quantified the percentage of hair follicle bulges with Tyrp1+ McSCs. Normally McSCs located in bulge area express DCT but not Tyrp1, which is a marker for differentiated melanocytes. So the ectopic Tyrp1+ McSCs are detected by Tyrp1 and DCT co-staining. We quantified hair follicle bulges with Tyrp1 positive staining signal in DCT+ cells and calculated their percentage. This has been explained more clearly in figure legends where the quantification applies.

2) The analysis of cell proliferation should be improved. The authors analyze Ki67+ cells but melanocyte stem cells may not express Ki67. The results could be explained by proliferation of melanocyte stem cells and by increasing numbers, more cells differentiate.

We agree with the reviewers that McSC proliferation should be carefully examined. In addition to Ki67 staining we also used quantification of McSC number as a direct readout of whether or not more McSCs are present in the niche under different conditions. Since we did not detect any difference in McSC number we put all these negative results in the supplementary data: in Figure 2—figure supplement 1E, we used DCT staining to quantify the McSC number in WT and *Krt6CreER::RBPJ cKO HFs* and found no difference in cell numbers; in Figure 3—figure supplement 2K, we quantified the McSC number in WT and *Krt6CreER::RBPJ cKO HFs* treated with Wnt inhibitor LGK974 and found no difference in cell numbers; in Figure 6—figure supplement 1B, we quantified the McSC number in bulge and melanocytes number in bulb of WT and *Krt6CreER::RBPJ cKO HFs* treated with RA synthesis inhibitor WIN 18446 and found no difference in cell numbers; in Figure 7—figure supplement 1D, we quantified the McSC number in WT HFs treated with different dosage of RA and found no difference in cell numbers. So based on these results we excluded proliferation as the cause of abnormal differentiation.

3) Does RA increase cKit expression or the number of ckit high cells? This part of the manuscript is not clear to me and seems essential for the proposed model.

Thank you for pointing this out. Based on our quantification of MsSC number using DCT staining described above, we did not detect increase in cell number. This is confirmed using FACS quantification of the c-Kit+ cells presented in Figure 6—figure supplement 1A. But the percentage of c-Kit^high^ among the c-Kit+ McSCs are changed after RA synthesis inhibtor WIN18446 treatment of the cKO HFs (Figure 6A). RA treatment increases c-Kit protein level (Figure 6B) and transforms a part of c-Kit+ but low cells into cKit high cells that we quantified using FACS (Figure 6A). So RA increases the c-Kit protein level and this results in increased number of c-Kit high cells, without change the overall number of McSCs. We have modified the description of this part in Results section to make the conclusion clear.

4) HFSCs are known to regulate McSCs via Wnt and Edn signaling. In addition, loss of Nfib, a transcription factor in HFSCs, showed a similar phenotype as the RBPj mutant described here. What are the similarities and differences between the RBPj mutant and some of these mutants that have been described before (in addition to RA being a metabolite)? The authors' story is a substantial departure from these previous papers regarding cellular and molecular mechanisms. This said, discussing some of these previous findings in greater depth will help the readers to appreciate the authors' findings in the right context and also help to differentiate the authors' findings from the previous ones even better. It might also be helpful to look into the authors' existing RNAseq data to see if Nfib/Edn signaling levels might be different in Rbpj mutant.

Thank you for these suggestions. In the Introduction we already included the Nfib/Wnt/Edn functions published previously. Now we have added more in depth comparison in the Discussion part. In our RNA-seq, Nfib expression is the same in Ctrl and cKO HFSCs (232.198 v.s. 203.073). In the Nfib mutants mice the melanocyte number is increased in bulge that is not observed in our RBP-J mutants. Edn family genes are either not expressed or shows decreased expression in cKO HFSCs. (Edn1: 0.237 v.s. 0.155; Edn2: 4.10 v.s. 1.75; Edn3: 0.073 v.s. 0). We have also functionally ruled out Wnt playing a role in the phenotype we observed. So based on these results we conclude the phenotype in our mutant is caused by novel mechanism different from previously published findings of how HFSCs regulate McSC functions. We have included these information in the Discussion part, which we agree will help the readers to appreciate our finding in the right context.

5) The manuscript can benefit from some trimming. The detailed descriptions of the data generated made the current manuscript quite complex and less accessible to readers. This is a really excellent work with strong data support for all the claims. Focusing on elaborating the main messages while going light on the description of supportive /non-essential data will help the story to stand out even more and will help the readers from diverse fields to appreciate the beauty of the story.

Thank you for making these suggestions. We have trimmed down a lot of the description about our negative result parts and elaborated more on the important and essential discoveries. Especially about the points emphasized above. Hopefully the readers will be able to follow our data more easily and appreciate our finding.